# The Role of lncRNAs in Complicated Pregnancy: A Systematic Review

**DOI:** 10.3390/genes16080959

**Published:** 2025-08-14

**Authors:** Antonio Cerillo, Rossella Molitierno, Pasquale De Franciscis, Debora Damiana Nunziata, Mario Fordellone, Carlo Capristo, Maria Maddalena Marrapodi, Andrea Etrusco, Antonio Simone Laganà, Marco La Verde

**Affiliations:** 1Department of Woman, Child and General and Specialized Surgery, University of Campania “Luigi Vanvitelli”, 80138 Naples, Italy; antoniocerillo@gmail.com (A.C.); molitiernorossella@gmail.com (R.M.); pasquale.defranciscis@unicampania.it (P.D.F.); deboradamiananunziata@gmail.com (D.D.N.); carlo.capristo@unicampania.it (C.C.); mariamaddalena.marrapodi@unicampania.it (M.M.M.); 2Medical Statistics Unit, University of Campania Luigi Vanvitelli, 80138 Naples, Italy; mario.fordellone@unicampania.it; 3Unit of Obstetrics and Gynecology, “Paolo Giaccone” Hospital, Department of Health Promotion, Mother and Child Care, Internal Medicine and Medical Specialties (PROMISE), University of Palermo, 90127 Palermo, Italy; etruscoandrea@gmail.com (A.E.); antoniosimone.lagana@unipa.it (A.S.L.)

**Keywords:** gestational diabetes mellitus, pregnancy-induced hypertension, gestational hypertension, pre-eclampsia, intrahepatic cholestasis of pregnancy, long non-coding RNA, lncRNA, biomarker

## Abstract

Background/Objectives: Long non-coding RNAs (lncRNAs) play a crucial role in trophoblast invasion, immune tolerance, and placental angiogenesis. To delineate their diagnostic and pathological significance, we critically evaluated the evidence for correlations between circulating or placental lncRNA profiles with pregnancy complications. Methods: Five databases were searched from inception through September 2024. We included only the studies that assessed the expression of the lncRNA-complicated pregnancies versus a control group. Results: Three single-center case–control studies fulfilled the inclusion criteria. Eight serum lncRNAs that present <20 weeks of gestation were elevated in subsequent pregnancy-induced hypertension or preeclampsia. The three lncRNAs in intrahepatic cholestasis of pregnancy were consistently decreased with a negative correlation with bile acids. Gestational diabetes was characterized by the elevation of MALAT1. Conclusions: Different lncRNAs showed a potential for use as non-invasive markers as well as for risk stratification for pregnancy-induced hypertension or preeclampsia, metabolic, and hepatobiliary pregnancy complications. There is a need for large-scale, multi-ethnic, prospective cohorts to include lncRNA as screening or therapeutic targeting in obstetric practice.

## 1. Introduction

The molecular basis of pregnancy involves complex interaction between physiological and pathological processes [1]. Any imbalance will result in adverse pregnancy outcomes (APOs), and, therefore, learning about molecular mediators in prenatal biology is critical [1]. Aberrant ncRNA expression in the placenta will disrupt trophoblast activities such as proliferation, invasion, and migration and result in APOs [2]. Long non-coding RNAs, or lncRNAs, are non-protein-coding RNA transcripts that are greater than 200 nucleotides in length [3]. lncRNAs play important roles in transcriptional regulation, impacting important processes such as differentiation, development, and response to disease [4]. lncRNAs can remodel the chromatin and alter gene access to transcription [5]. These molecules regulate transcription by interacting with transcription factors and with RNA polymerase II, either stimulating or inhibiting the expression of certain genes [6]. lncRNAs also regulate post-transcriptional processes by regulating the production of proteins [7]. Aberrant lncRNA expression is associated with various pathological conditions, including cancers, cardiovascular disease, and pregnancy complications [8,9,10]. Long non-coding RNA plays an important role in contributing to pregnancy and reproductive health [11]. Historically, the placenta is defined by the mother-to-fetus interface, with basic functions involving nutrient and gas exchange, waste removal, and immune tolerance [12,13,14]. Of particular interest is the ability to regulate the mother’s immune system to prevent rejection of the semi-allogeneic fetus [15]. The placenta achieves immune tolerance by mechanisms involving the secretion of immunomodulatory factors, non-classical MHC molecule expression, and maintenance of a physical barrier [14,16]. Non-coding RNAs have been identified to be critical regulators in these processes in recent times [17]. Deregulation of lncRNAs, including PART1, is identified to be associated with abnormal trophoblast proliferation and invasion and impaired interaction with maternal immune cells in the fetal–maternal interface [18]. Experimental evidence also confirms the involvement of PART1 in pregnancy complications including unexplained recurrent pregnancy loss (URPL) and preeclampsia complications [2]. Another long non-coding RNA, GAS5, has been identified as a regulator of immune responses and is associated with autoimmune diseases, though its role in RPL pathophysiology is unclear [19]. The molecular processes in APOs are incomplete. In pregnancy complications, although there is evidence showing the upregulation of lncRNAs’ expression, the functional implication of their dysregulation is unknown. Moreover, the processes by which it acts to regulate placental biology and as a therapeutic target are unknown. Early evidence suggests that certain lncRNAs may play a hub function in signaling pathways that regulate placental development [11,20]. Their interaction with microRNAs (miRNAs) and transcription factors is said to mediate immune modulation and trophoblast differentiation [21]. These pathways and their role in pregnancy outcomes, though, are unmapped. The aim of this systematic review is to explore the relationship between lncRNAs and adverse pregnancy outcomes. In particular, we want to describe lncRNA expression patterns under pathological pregnancy, to establish their possible clinical relevance as a pregnancy diagnostic marker and therapeutic target. These results could promise to bring about a new era in pregnancy biology and better clinical practice.

## 2. Methods

### 2.1. Search Strategy

Literature searches of Medline PubMed, EMBASE, Scopus, the Cochrane Library, and Research Register (ClinicalTrial.gov) databases were conducted from the outset to September 2024 with the search terms “lncRNA” and “ pregnancy outcomes”. The following search strategies were developed using MeSH (Medical Subject Headings) terms and Boolean operators to search Medline database: (“pregnancy outcome” [MeSH Terms] OR (“pregnancy” [All Fields] AND “outcome” [All Fields]) OR “pregnancy outcome” [All Fields]) AND (“lncrnas” [All Fields] OR “RNA, long noncoding” [MeSH Terms] OR (“RNA” [All Fields] AND “long” [All Fields] AND “noncoding” [All Fields]) OR “long noncoding RNA” [All Fields] OR “lncrna” [All Fields]). Titles and abstracts were independently screened, and the full-text articles were assessed by two reviewers (DDN and AC). Further articles were identified by scanning the reference lists of included studies. Search results and the included articles were screened by a third reviewer (RM). Differences were resolved by consensus. Review methods were established at the outset in accordance with Preferred Reporting Items for Systematic Reviews and Meta-Analyses [22]. Review registration on the PROSPERO site had the number CRD420251027090.The PRISMA 2020 Checklist was added in the Appendix A. 

### 2.2. Eligibility Criteria

The pre-defined inclusion criteria targeted studies involving pregnant women and the link between the pregnancy outcome and the lncRNA. Review outcomes were the lncRNA expression in complicated pregnancy. No studies were excluded due to language and geographic location to capture the maximum comprehensive range of findings relevant to the review question. Excluded studies were review articles, meta-analyses, and those in which the study did not involve lncRNAs or pregnancy complications.

### 2.3. Data Extraction and Analysis

Extracted information included population features, lncRNAs, and pregnancy outcomes. Data extraction was performed by two reviewers (DDN and AC), and data accuracy and completeness were checked by the third reviewer (RM). Because of substantive heterogeneity in study methods and heterogeneity in data reported, abnormalities investigated, and population features, it was not suitable to undertake a quantitative analysis. All the included studies were assessed for possible conflicts of interest.

### 2.4. Methodological Quality Assessment

We assessed the quality of the included studies based on a modified Newcastle–Ottawa scale (NOS) [23], with five separate domains outlined in Table 1.

Quality of the study was independently assessed by two authors (DDN and AC). Any disagreement was resolved by discussion or by reference to RM.

### 2.5. Outcome Measures

The primary objective of this review is to assess the expression of the lncRNA (Intervention) in the pregnant women (Population). lncRNA expression was compared between healthy pregnancies and complicated pregnancies (Comparison). Specifically, the association of lncRNA with pregnancy outcomes will be evaluated (Outcome).

## 3. Results

### 3.1. Study Selection

The process of study selection is outlined in Figure 1.

In total, 205 papers were initially identified; 37 duplicates were removed. The remaining 168 articles underwent title and abstract screening, during which 123 records were excluded based on predefined criteria (e.g., not related to pregnancy complications or lncRNAs, non-original studies, language restrictions). In total, 45 articles were selected for full-text review, and a total of 3 papers [24,25,26] fulfilled the inclusion criteria and were selected in the current systematic review.

### 3.2. Study Characteristics

The principal characteristics of the included studies are shown in Table 2.

The studies were all carried out in China in a single center using a case–control design. The studies were retrospective in one study [26] and nested case–control [24] or routine case–control [25]. The studies included lncRNA profiling and biomarker measurements [24], microarray screening confirmed by qRT-PCR [25], and a sampling of sera and measurement by qRT-PCR [26]. Sample sizes ranged between 97 and 204 subjects. Dai et al. explored preeclampsia (PE) and pregnancy-induced hypertension (PIH) and included a screening group (n = 10) and a validation group (n = 204 equally divided between PIH and normotensive pregnancy) [24]. Zou et al. analyzed the intrahepatic cholestasis (ICP) and analyzed a total number of 108 women, 54 with ICP and 54 with a low-risk pregnancy [25]. Zhang et al. enrolled 97 women with 50 cases of gestational diabetes mellitus (GDM) and 47 controls [26]. Gestational age included the third trimester of pregnancy. Maternal age ranged between 20 and 35 years.

### 3.3. Risk of Bias of Included Studies

All three studies included in this review had a low risk of bias in three or more domains [24,25,26]. Dai et al. presented five out of five domains with a low risk [24]. An analysis of individual risk of bias by domain is reported in Table 3.

### 3.4. Synthesis of Results

The three included studies explored all three different maternal comorbidities. Dai et al. explored the PE/PIH [24], Zou et al. explored the ICP [25], and Zhang et al. explored the GDM [26] (Table 4).

Following, we reported the results of the different pathologies explored by each study.

#### 3.4.1. Hypertensive Disorders of Pregnancy

Dai et al. employed a nested case–control design to study a set of lncRNAs for their association with PIH and PE [24]. Eight lncRNAs were elevated in PIH compared to normotensive controls, and seven of them were also found to have increased expression in PE cases. Specifically, LINC03097 (ENST00000527727) appeared associated with fetal growth restriction (FGR), and H19 (ENST00000415029) with placenta accreta. Early sampling (before week 20 of gestation) was adopted in this study, and final delivery occurred at week 39. All lncRNAs exhibited AUC values >0.6 and therefore good diagnostic utility in early prediction and stratification of risk.

#### 3.4.2. Intrahepatic Cholestasis of Pregnancy

Three lncRNAs, ENST00000505175.1, ASO3480, and ENST00000449605.1, resulted in downregulated in ICP patients. The lncRNAs were found to have an inverse correlation with total bile acids and liver function indices. AUC ranged between 0.731 and 0.812, with a combined AUC value of 0.865. The decreased expression levels were also found to be associated with adverse perinatal outcomes in regard to meconium-stained amniotic fluid, fetal distress, and prematurity and are thus effective for other biomarkers in clinical application [24].

#### 3.4.3. Gestational Diabetes Mellitus

Zhang et al. explored MALAT1 expression and lncRNA p21 and lncRNA H19 in patients with GDM. MALAT1 showed a significant expression in GDM patients versus the controls. lncRNA p21 and H19 showed no differences between the two groups. MALAT1 exhibited an AUC = 0.654 with ~50% sensitivity and ~83% [26].

## 4. Discussion

PIH/PE, ICP, and GMD were explored by our review. The three case–control studies, included to explore the lncRNA relation with the obstetric complications, evidenced a lncRNA role for diagnosis and prognosis [24,25,26]. Of these lncRNAs, eight were found by Dai et al. to be induced in hypertensive pregnancy [24]. In the study conducted by Dai et al., lncRNA expression was measured in early pregnancy (<20 weeks’ gestation) to determine its predictive capability for PIH, including PE. Eight candidate lncRNAs were identified to be significantly upregulated in women who developed PE/gestational hypertension. Prominently, ENST00000527727 and ENST00000415029 were correlated to adverse outcomes like FGR and placenta accreta, respectively [24]. Functional enrichment analysis associated these lncRNAs with processes such as MAPK signaling and lipid metabolism, hinting at an involvement in early pathological changes leading to PE [24]. This suggests a possible role for lncRNAs as early severity markers for PIH/PE and to have correlations with complications such as FGR and placenta accreta [24]. Zou et al. explored the third-trimester ICP women [25]. Three lncRNAs, ENST00000505175.1, ASO3480, and ENST00000449605.1, were significantly downregulated in ICP patients. They were positively correlated with total bile acids and had high diagnostic values according to the ROC analysis (AUC > 0.8 in combination) [25].

Zhang et al. explored the levels of three particular lncRNAs, MALAT1, H19, and p21, in maternal serum at term in GDM pregnant women [26]. MALAT1 was much more highly expressed in GDMs compared to the controls and positively associated with lipid profiles and other lncRNAs (H19 and p21), suggesting a synchronized regulatory role in the metabolic dysfunction. Although H19 and p21 did not display dysregulation individually, their co-expression with MALAT1 suggests possible roles in disease progression [26].

The scientific literature demonstrated the lncRNA role in pregnancy outcome prediction in unexplained recurrent pregnancy loss (URPL) [2]. lncRNA is present in high levels in the endometrial tissues of URPL patients and may be a potential biomarker and therapy target [2]. Non-coding RNAs (ncRNAs) play a regulatory role in most cells and tissues and are highlighted by genome programs such as ENCODE [27]. Most of the known placental lncRNAs reported to date are miRNAs [28]. Syncytiotrophoblast releases extracellular vesicles into the maternal blood with fetal RNA and protein, including ncRNAs, to convey information to maternal cells [29]. Aberrations in ncRNA in the placenta and EVs have been implicated in pregnancy disorders [28]. This finding is in accord with the general understanding of the major functions of non-coding RNAs in the regulation of reproductive functions.

Majevska et al. identified several isoforms of different annotated loci of lncRNAs [30]. Placental lncRNAs were shorter in transcript length, had larger exons, fewer exons, and were less expressed compared to mRNAs [31]. Different expressing lncRNAs regarding fetal sex and splicing activities were identified [32,33,34]. These findings contribute to the understanding of lncRNA in the placenta and enhance human non-coding RNA catalogs [30]. The upregulation of particular lncRNAs, including PART1 in URPL cases, suggests its role in the pathophysiology of recurrent pregnancy loss [2]. Also, the roles of lncRNAs in modulating placental trophoblast invasion into endometrial stromal cells have been documented and are significant in facilitating or inhibiting the process (Table 5) [28]. These results not only validate lncRNA as a potential biomarker but also as a potential target for use in therapy [35]. These findings validate the general paradigm that non-coding RNAs have significant regulatory functions in reproductive processes [36]. Overall, the evidence suggests the importance of lncRNA to reproductive health and their possible clinical applications to improve pregnancy. Dynamic fluctuations in the mother serum level of exosomal lncRNA have the potential to be a reliable and timely marker of pregnancy-related disease development and recovery and are an attractive route to the development of clinical diagnostics and therapeutics (Table 5) [37]. This could be attributed partly to the general prediction of up- and downregulation of lncRNA for various pathological conditions as it is established to bind mRNAs as well as miRNAs [37]. However, the striking relationship between some of the lncRNA expressions and pregnancy outcomes, such as FGR and even miscarriage rates, was significantly greater than expected based on comparison with better-characterized classes of RNAs enabling modulation of maternal–fetal communication in normal and abnormal pregnancy (Table 5). These findings reflect the striking implication of lncRNAs in the fine-tuning of maternal–fetal communication.

Our results concur with the wider literature on lncRNAs and pregnancy function. Long non-coding RNAs have been discovered in recent years to play significant functions in trophoblast invasion, decidualization, and maternal–fetal interface immune tolerance. On the other hand, the regulatory function of lncRNAs is emphasized in endometrial stromal cells and trophoblast differentiation in implantation, inflammation, and subsequent placentation [38,39,40]. Preeclampsia pathogenesis is associated with abnormal placentation and altered maternal vascular responses. Two other studies supported that lncRNAs and circulating cell-free RNAs (cfRNAs) have possible preeclampsia biomarkers. In a prospective large cohort, Zhou et al. used polyadenylation ligation-mediated sequencing (PALM-Seq) to provide a comprehensive view of plasma cfRNA across 917 pregnancies, including 202 pregnancies that developed preterm or early-onset preeclampsia [41]. Their integrative model using 13 cfRNA biomarkers and the clinical characteristics (mean arterial pressure and IVF conception) exhibited a stable predictive capability (AUC 0.81–0.88). Specifically, a number of downregulated microRNAs and lncRNAs were linked to derepression of gene networks involved in endothelial dysfunction [42]. Complementarily, Ogoyama et al. performed a review on circulating and placental non-coding RNAs highlighting the putative impact of these RNAs on trophoblast invasion and vascular remodeling during early gestation. They reported a number of placenta-related lncRNAs, including H19 and miR-210, which were concomitantly increased in early placenta and maternal blood, supporting their identification as early biomarkers of prediction and as potential mediators in placental maladaptation. H19 was expressed during the first trimester of pregnancy in women who later develop preeclampsia. This review highlighted the roles of H19 and miR-210 in early placenta alterations and maternal blood, which supports their identification as early biomarkers of placental maladaptation.

There are several strengths in this systematic review. This systematic review was the first that focused on the pregnancy outcomes and their relations with the lncRNA. This review was conducted according to a pre-registered protocol and involved a comprehensive search strategy in a selection of major databases to improve methodology transparency [22]. The inclusion criteria were non-selective in terms of geography or language to contribute to the generality of the evidence base, and studies that reported only lncRNA were included. Quality was evaluated through a modified Newcastle–Ottawa Scale applied by two independent reviewers [23]. Nevertheless, this review had different limitations. First, only three studies met the inclusion criteria. All studies were derived from single-center studies in China potentially affecting external validity and generalizability of conclusions. In addition, the possible sharing of the control group in the involved studies from the same hospital creates a risk of sample duplicity, possibly affecting the identified associations and may restrict their interpretation in comparison between studies. Different maternal outcomes were explored; consequently, the quantitative analysis was not feasible. Despite these limitations, findings validate the potential clinical utility of lncRNAs in pregnancy monitoring and stimulate future research.

Further studies are required to determine the specific pathways by which lncRNA plays a role in pregnancy. Large-scale cohort studies have to be performed in order to confirm the diagnostic value of lncRNA and develop reference values to facilitate clinical use. In addition, future investigations should explore the lncRNA diagnostic prediction for pregnancy complications. In particular, multivariate analyses should be used to evaluate the independent impact of individual lncRNAs on adverse pregnancy outcomes after controlling for possible confounders. These studies can also explore the relationship between lncRNA, other lncRNAs, and their targets and clarify the larger pregnancy-controlling pathways.

## 5. Conclusions

LncRNA could be considered a possible regulator of pregnancy outcomes and potentially could contribute to the diagnosis and treatment of PIH/PE, ICP, and GDM. Considering the dual function as a drug target and as a biomarker, lncRNA could play a significant role in future innovations in maternal–fetal medicine. Further study into the molecular mechanism of lncRNA and its clinical application will be required to expand the clinical role of lncRNAs.

## Figures and Tables

**Figure 1 genes-16-00959-f001:**
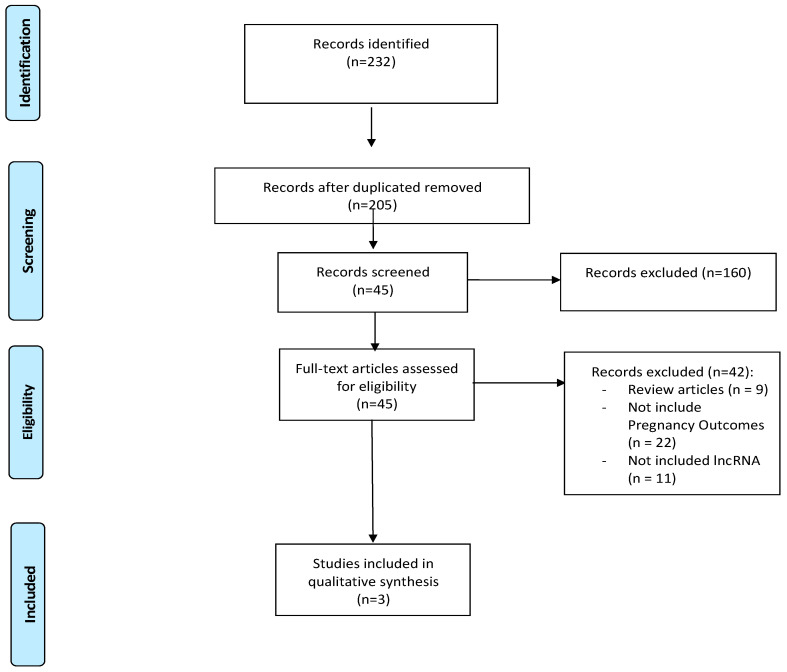
PRISMA flow diagram of this review.

**Table 1 genes-16-00959-t001:** Modified Newcastle–Ottawa scoring items.

(1) Study design and sample representativeness:
1 point: Study includes a sample size greater than or equal to 100.
0 points: Sample size of less than 100 participants.
(2) Sampling technique:
1 point: Patients recruited consecutively or randomly (randomization criteria clarified).
0 points: Potential convenience sampling or unspecified sampling technique.
(3) Description of the lncRNA technique:
1 point: The authors provided a comprehensive description of the equipment, setting, and adopted technique.
0 points: The study did not report adequate information on the lncRNA technique.
(4) Quality of population description:
1 point: The study reported a clear description of the population with proper measures of dispersion (e.g., mean, standard deviation).
0 points: The study did not report a clear description of the population, incompletely reported descriptive statistics or did not report measures of dispersion.
(5) Incomplete outcome data:
1 point: The study reported complete data about the pregnancy outcomes.
0 points: Selective data reporting cannot be excluded.

The individual components listed above are summed to generate a total modified Newcastle–Ottawa risk of bias score for each study. Total scores range from 0 to 5. For the total score grouping, studies are judged to be of low risk of bias (≥3 points) or high risk of bias (<3 points).

**Table 2 genes-16-00959-t002:** Characteristics of the included studies and study design/methodology.

Study (Year)	Country	Study Design/ Setting	Methodology	Population and Sample Size	Gestational Age (GA)	Maternal Age
Dai et al. (2021) [24]	China	- Nested case–control study - Single-center	- lncRNA profiling + biomarker study	- 10 total (5 PE and 5 matched controls) in the screening stage - 102 women with PIH and 102 normotensive (validation phase)	- Sampling less than 20 weeks of gestation for early detection - Latest delivery data examined through ~39 weeks	20–35 years
Zou et al. (2021) [25]	China	- Case–control study - Single center	- Microarray screening + qRT-PCR validation	- 54 with intrahepatic cholestasis of pregnancy (ICP) - 54 healthy pregnant women	- In late 3rd trimester (average ~37.9 weeks in ICP versus ~38.9 weeks in controls)	26–29 years
Zhang et al. (2018) [26]	China	- Retrospective case–control study - Single-center	- Serum sampling + qRT-PCR	- 50 women with gestational diabetes mellitus (GDM) - 47 controls	36–40 weeks of gestation	28–32 years

**Table 3 genes-16-00959-t003:** Risk of bias assessment of the 3 included studies.

Author, Year	Study Design and Sample Representativeness	Sampling Technique	Description of the lncRNA Technique	Quality of Population Description	Incomplete Outcome Data	Total Score
Dai et al. (2021) [24]	★	★	★	★	★	★★★★★
Zou et al. (2021) [25]	★	-	★	★	★	★★★★
Zhang et al. (2018) [26]	-	-	★	★	★	★★★

**Table 4 genes-16-00959-t004:** Principal lncRNAs examined, expression patterns, and relationship with obstetric outcomes.

Study	Pregnancy Complication Evaluated	Investigated lncRNAs	Expression/Changes	Correlation with Outcomes/Key Findings	Ref.
Dai et al. (2021)	- Gestational Hypertension or Preeclampsia	- *NR_002187, ENST00000415029, ENST00000398554, ENST00000586560, TCONS_00008014, ENST00000546789, ENST00000610270, ENST00000527727*	- All upregulated in GH/PE, compared with normotensive controls - 7/8 of these (all but *ENST00000415029*) had even greater levels in PE cases, which implies association with greater disease severity	- Early Potential for diagnosis: AUC > 0.6 for all lncRNAs - Some are implicated in unfavorable outcomes (e.g., higher *ENST00000527727* correlates with FGR; *ENST00000415029* linked to placenta accreta) - Can assist in risk stratification and proactive surveillance	[24]
Zou et al. (2021)	- Intrahepatic Cholestasis Pregnancy (ICP)	- *ENST00000505175.1, ASO3480, ENST00000449605.1* (all downregulated)	- *ENST00000505175.1, ASO3480, and ENST00000449605.1* are significantly decreased in ICP patients’ serum compared with controls - Highly significant correlations with bile acid levels (TBA) and liver function indices	- Diagnostic potential: AUC from 0.731 to 0.812 for each lncRNA; combined AUC = 0.865 - Prognostic significance: decreased levels of these lncRNAs correlating with adverse perinatal outcomes (meconium-stained amniotic fluid, fetal distress, and preterm birth) - They suggest employing them as adjunctive biomarkers in combination with TBA for enhanced risk identification and management in ICP.	[25]
Zhang et al. (2018)	Gestational Diabetes Mellitus (GDM)	- *MALAT1* (primary) - *lncRNA p21* - *lncRNA H19*	- *MALAT1* is increased in GDM patients - Positive correlation between p21 and H19, but p21 and H19 themselves are not different significantly between GDM compared to controls	- *MALAT 1* is associated with glucose/lipid metabolism. *MALAT1* is a potential diagnostic biomarker of GDM (AUC = 0.654); ~50% sensitivity, ~83% specificity transcription factor stability, leading to GDM pathogenesis	[26]

**Table 5 genes-16-00959-t005:** Strengths and limitations of lncRNAs.

lncRNAs	Strengths	Limitations
Biological relevance	lncRNAs are involved in reproductive processes like trophoblast invasion, decidualization, and maternal immune tolerance.	The molecular lncRNAs’ pathways is incompletely mapped, and clinical adoption is limited.
Diagnostic and prognostic value	lncRNAs in maternal serum represent a promising, non-invasive way for early screening of pregnancy complications.	Actually, a robust diagnostic cutoff is lacking.
Clinical potential	Several lncRNAs show associations with adverse outcomes.	Validation is still lacking in large, multi-ethnic, prospective cohorts.
Mechanistic plausibility	lncRNAs interact with both mRNAs and miRNAs to regulate gene expression at transcriptional, epigenetic, and post-transcriptional levels.	The complexity and redundancy of lncRNAs make it difficult to isolate individual contributions.

## Data Availability

All data generated or analyzed during this study are included in this article.

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
