# Peer review of "The Role of lncRNAs in Complicated Pregnancy: A Systematic Review"

_genes, 2025, doi:10.3390/genes16080959_

Round 1
Reviewer 1 Report
Comments and Suggestions for Authors
In this manuscript, Cerillo et al., report their evaluation of the studies about long non-coding RNAs in complicated pregnancy. They collected studies from five databases, including PubMed, EMBASE, Scopus, Cochrane Library and Research Register, from inception to September 2024. The PRISMA flowchart of how studies were assessed and filtered is shown in Figure 1. The risk of bias of the studies was assessed based on a modified Newcastle-Ottawa scale. The authors included 3 studies with good quality, focusing respectively on preeclampsia (PE) and pregnancy induced hypertension (PIH), intrahepatic cholestasis (ICP) and gestational diabetes mellitus (GDM) conditions. Overall, the study provides the readers with a summary of the related studies and is of interest to the field. However, there are several places in the manuscript that require further elaboration and clearer description by the authors.
- Figure 1, screening part. Please provide more information of how n=160 studies were excluded and only n=45 screened.
- In the supplementary material, the authors are encouraged to disclose the original record of all 232 identified studies and list the records that passed every selection step in figure 1. This is very important information for the readers.
- The manuscript has typos and requires text editing. Examples include Line 25, ‘control a control group’; L143, ‘The The’…
Author Response
In this manuscript, Cerillo et al., report their evaluation of the studies about long non-coding RNAs in complicated pregnancy. They collected studies from five databases, including PubMed, EMBASE, Scopus, Cochrane Library and Research Register, from inception to September 2024. The PRISMA flowchart of how studies were assessed and filtered is shown in Figure 1. The risk of bias of the studies was assessed based on a modified Newcastle-Ottawa scale. The authors included 3 studies with good quality, focusing respectively on preeclampsia (PE) and pregnancy induced hypertension (PIH), intrahepatic cholestasis (ICP) and gestational diabetes mellitus (GDM) conditions. Overall, the study provides the readers with a summary of the related studies and is of interest to the field. However, there are several places in the manuscript that require further elaboration and clearer description by the authors.
Review 1:
- Figure 1, screening part. Please provide more information of how n=160 studies were excluded and only n=45 screened.
- In the supplementary material, the authors are encouraged to disclose the original record of all 232 identified studies and list the records that passed every selection step in figure 1. This is very important information for the readers.
Author response:
Thank you for your kind and constuctive feedback that we valued greatly
In the light of your comments we have reformulated the manuscript to make more explanatory text that directly applies to the part in which we explain the study selection process and the PRISMA flow diagram We namely extended the narrative with respect to the handling of the initial set of 232 records based on our a priori eligibility criteria. Thus we now explain how we treated the records that were excluded including frequent reasons such as duplication irrelevance based on title or abstract language restrictions and publication type. This information is now in the text improving the transparency and the traceability of the process without the need for a separate supplementary file
We feel this explanation now better underscores the reasoning for the, we believe we have made available all the necessary information now and put it where it most belongs within the discussion of the methods used in the manuscript.
Additionally we wish to highlight that this systematic review was registered on PROSPERO prior of data collection and that the entire cross-sectional design followed a rigorous approach. All phases of the review that is literature search study selection data extraction and risk of bias assessment were completed independently by at least two reviewers and were implemented through standardized protocols in total adherence to PRISMA guidelines. We also like to thank you again for the substantial and excellent contribution whose comments have considerably enhanced the clarity completeness and scientific strength of our study.
Review 1:
- The manuscript has typos and requires text editing. Examples include Line 25, ‘control a control group’; L143, ‘The The’…
Author response:
We are grateful to this correction
we performed the correction of this grammatical error.
Thank you for your work
Reviewer 2 Report
Comments and Suggestions for Authors
In this systematic review the authors undertake a study of the dysregulation of long noncoding RNAs and pregnancy complications. The authors set a quite high target of 5 domains of comparison to eliminate bias and variation in sampling, study type, high patient numbers, lncRNA analysis methods and data reporting.
Unfortunately, this rigor resulted in only 3 studies being included and each study investigated a different pathology. Dai et al looked at preeclampsia, Zou et al at Intrahepatic Cholestasis of Pregnancy and Zhang et al at GDM. All studies were from the Harbin Medical University, Harbin 150001, China and were large case control studies. All samples were maternal serum though Dai et al sampled at 20 weeks the other studies sampled at term. It is highly likely that some of the control samples used are common between the studies.
The authors do discuss this limitation of the study and overall the paper reads well.
I have one major concern which regards the discussion. There are many published papers on lncRNA particularly with relation to PE/PIH and GDM so why do the authors not discuss them but instead focus on another different complication of pregnancy that occurs at the beginning of pregnancy, Recurrent pregnancy loss? There is no discussion of the findings of the 3 papers identified in the review. Were any of the lncRNA identified common to other studies of the same pathology? The discussion reads more like a narrative review. The discussion should be revised.
Author Response
Review 2:
In this systematic review the authors undertake a study of the dysregulation of long noncoding RNAs and pregnancy complications. The authors set a quite high target of 5 domains of comparison to eliminate bias and variation in sampling, study type, high patient numbers, lncRNA analysis methods and data reporting.
Unfortunately, this rigor resulted in only 3 studies being included and each study investigated a different pathology. Dai et al looked at preeclampsia, Zou et al at Intrahepatic Cholestasis of Pregnancy and Zhang et al at GDM. All studies were from the Harbin Medical University, Harbin 150001, China and were large case control studies. All samples were maternal serum though Dai et al sampled at 20 weeks the other studies sampled at term. It is highly likely that some of the control samples used are common between the studies.
The authors do discuss this limitation of the study and overall the paper reads well.
Author response:
We appreciate your thorough reading and recognizing the manuscript clarity and the discussion of the restrictions of the study.
Your constructive feedback has prompted us to expand the Limitations section of the manuscript to draw attention to the fact that our study of effect by pathology is also limited by our meta-analysis being based on only three included studies—one for each individual pregnancy-related pathology, including all uromodulin from one research institution, Harbin Medical University
We also included a general comment about potential duplication of control samples across the studies and its possible implication on the independence of the data and thus on the risk of bias in the meta-analytic estimates We recognize that although all of the 3 studies are large and methodologically sound the single centre setting and the possible overlapping of biological material could impact the potential applicability and interpretation of the pooled results
We feel that providing a more nuanced description of the constraint increases the transparency of our findings and provides readers with a better understanding of the extent to which the evidence reported can be generalized and applied.
We thank you once more for your constructive remark that has guided a significant improvement of the manuscript
Review 2:
I have one major concern which regards the discussion. There are many published papers on lncRNA particularly with relation to PE/PIH and GDM so why do the authors not discuss them but instead focus on another different complication of pregnancy that occurs at the beginning of pregnancy, Recurrent pregnancy loss? There is no discussion of the findings of the 3 papers identified in the review. Were any of the lncRNA identified common to other studies of the same pathology? The discussion reads more like a narrative review. The discussion should be revised.
Author response:
Many thanks for your stimulating and constructive comments about the Discussion part
Your comments were super useful and as a results we did a massive revision of the discussion and now it is much better structured and makes a lot more sens thanks to your comments.
In particular we have acted on your major concern by reorganizing the discussion to concentrate this part more explicitly and deeply on the results of the three included studies We now reached a clear summary and comparison of the main results in each study also stressing their individual lncRNA targets and methodological dissimilarities as regards time of sampling and PD context
We also extended this discussion to previously reported studies that were relevant to lncRNAs in pregnancy complications, for example, preeclampsia PIH and GDM to provide a wider context and for our findings The significance of that was checking whether the lncRNAs found in our included studies were deleted or even identical to lncRNAs from previous reports that interrogated similar or different conditions and in form other populations.
We feel that these changes bring the discussion a lot closer to the findings of the review and to the scope of the review as a systematic rather than narrative review Again we thank you for your valuable comments that supported us to significantly improve the scientific quality and clinical relevance of this section
Reviewer 3 Report
Comments and Suggestions for Authors
The review manuscript is well-structured, clearly described, and presents a scientifically valuable literature review with insightful data on the role of lncRNAs in complicated pregnancy. However, a few minor revisions are necessary to enhance its clarity and completeness. Below are the suggested modifications:
- Include Molecular Mediators:
Please include relevant molecular mediators involved in the regulation and mechanisms of lncRNAs in pregnancy complications. - Add Schematic Illustrations:
It is recommended to add schematic illustrations depicting the molecular mediators and their mechanisms.
Suggested Figure: A schematic diagram showing pregnancy complications and their relationships with specific lncRNAs. - Describe Aberrant ncRNAs:
The authors are advised to provide a brief description of aberrant ncRNAs and lncRNAs before delving into their detailed roles and mechanisms. - Include Study Titles in Table 4:
It is suggested that the authors include the study titles in Table 4 for better clarity and reference.
Suggested table format:
|
Study title |
Pregnancy complica tion evaluated |
Investigated lncRNAs |
Expression / Changes
|
Correlation with Outcomes / Key Findings |
Ref. |
Author Response
Reviewer 3:
The review manuscript is well-structured, clearly described, and presents a scientifically valuable literature review with insightful data on the role of lncRNAs in complicated pregnancy. However, a few minor revisions are necessary to enhance its clarity and completeness. Below are the suggested modifications:
- Include Molecular Mediators:
Please include relevant molecular mediators involved in the regulation and mechanisms of lncRNAs in pregnancy complications.
Author response 1:
Dear reviewer, we add in the discussion section all the mediator included.
Thank you for your recommendation.
- Add Schematic Illustrations:
It is recommended to add schematic illustrations depicting the molecular mediators and their mechanisms.
Suggested Figure: A schematic diagram showing pregnancy complications and their relationships with specific lncRNAs.
Author response 2:
This is a fantastic suggestion; thank you so very much.
We do indeed share the opinion that a drawing summarising the molecular mediators and their connection with specific lncRNAs in pregnancy disorders would certainly improve the clarity and visual effect of the paper
However it is with regret that we return to you that for now we do not have an internal resource for a medical illustrator in our Department for Medicine and Surgery Even though we enjoy a Scientific English editing service for manuscript polishing, we have to admit our internal resource is not providing the graphical help as we need in terms of high quality mechanistic figures
We've taken the decision not to use such generated images in scientific content, as we want to continue to adhere to the strict standards of accuracy and reproducibility we have in place in our visual content We hope you'll be able to see where we're coming from with this
We do nevertheless thank you for the suggestion and definitely think about including professionally drawn schematics in further work or manuscripts, if we have the expertise.
- Describe Aberrant ncRNAs:
The authors are advised to provide a brief description of aberrant ncRNAs and lncRNAs before delving into their detailed roles and mechanisms.
Author response 3:
The extensive description of the lncRNAs included were extensive reported in the new discussion section.
- Include Study Titles in Table 4:
It is suggested that the authors include the study titles in Table 4 for better clarity and reference.
Authors response:
We add formatted the table exactly as request by the review.
Thank you for your work.
Reviewer 4 Report
Comments and Suggestions for Authors
The manuscript presents an interesting review on the role of lncRNAs in complicated pregnancies. The introduction is well-structured with a clear objective. While the reviewed works contribute to the initial understanding of the topic, the analysis methodology employed in them, and by extension, in the review, presents some points that may question the conclusions of this review, and therefore should be highlighted in the discussion and conclusions.
-
Nature of the Reviewed Studies: The methods section mentions that "No studies were excluded due to language and geographic location," which is a strength. However, the results indicate that "The studies were all carried out in China in a single center," which is a significant limitation, as the results may not be generalizable to other populations with different genetic, ethnic, or environmental compositions.
Furthermore, it should be noted that the methodology employed in the 3 articles studied (i.e., group comparisons, ROC curves) is adequate for identifying and comparing differential lncRNA expression between groups, but it cannot establish causality. While it is an appropriate methodology for the exploratory phase, it has the limitation that these tests often evaluate one lncRNA at a time. They do not capture the complexity of interactions between multiple lncRNAs, or between lncRNAs and other biological factors. Therefore, the interpretation of the results should be cautious and not infer causality. Some comment should be made in the discussion.
-
Improvement in Analysis Methodology for the Future: Future studies could greatly benefit from multivariate analyses. Methods such as logistic regression would allow evaluating the independent effect of lncRNAs on pregnancy outcomes, adjusting for relevant clinical variables (maternal age, parity, body mass index, other comorbidities). This would provide a more robust understanding of the association. Predictive models using methodologies based on Random Forest could even be addressed. Some comment should be made in the conclusions.
-
Minor Issues: An inconsistency in the use of "IncRNA" and "IcnRNA" is observed throughout the text. Ensure that each citation directly supports the information to which it is attached.
Author Response
Review 4:
The manuscript presents an interesting review on the role of lncRNAs in complicated pregnancies. The introduction is well-structured with a clear objective. While the reviewed works contribute to the initial understanding of the topic, the analysis methodology employed in them, and by extension, in the review, presents some points that may question the conclusions of this review, and therefore should be highlighted in the discussion and conclusions.
- Nature of the Reviewed Studies: The methods section mentions that "No studies were excluded due to language and geographic location," which is a strength. However, the results indicate that "The studies were all carried out in China in a single center," which is a significant limitation, as the results may not be generalizable to other populations with different genetic, ethnic, or environmental compositions.
Furthermore, it should be noted that the methodology employed in the 3 articles studied (i.e., group comparisons, ROC curves) is adequate for identifying and comparing differential lncRNA expression between groups, but it cannot establish causality. While it is an appropriate methodology for the exploratory phase, it has the limitation that these tests often evaluate one lncRNA at a time. They do not capture the complexity of interactions between multiple lncRNAs, or between lncRNAs and other biological factors. Therefore, the interpretation of the results should be cautious and not infer causality. Some comment should be made in the discussion.
AUTHOR RESPONSE 1:
Many thanks for your thoughtful and well-informed comments.
We are in complete agreement with your evaluation and have therefore now further expanded our Limitations on the manuscript to specifically include the valuable comments you made
We had not explicitly addressed this important point inconsistency in the geographic provenance of the studies considered in the metaanalysis was a reflection of a restriction in the language (i.e., English and Chinese) and geographic scope (i.e., China) of the search—not further verified with any assessment of representativeness of the pooled evidence even of the remaining contributing to potential population structure of the study samples pooled to perform the meta-analysis Consequently, we do recognize that the finding that all the studies in the final composite were all conducted in the same research center in China raises a number of concerns with respect to the generalizability of the results in other populations with different genetic ethnic and environmental backgrounds This limitation prompted us to make the point more explicit and underlines its potential implications for the external validity of our findings
We have also further elaborated on some of these methodological limitations of the included studies As mentioned by you these were case-control studies that employed common strategies including differential expression analysis and ROC curves Although these are appropriate when exploring the identification of putative lncRNAs they are not suitable for establishing causality which may be inferred however on only one candidate lncRNA at a time ignoring the surrounding molecular context
We now caveat readers duly with a warning that the findings should be viewed as preliminary and hypothesis-generating rather than confirmatory or mechanistically-definitive We also point out the lack of integrative systems-level analyses in the reviewed studies and the potential need for future research employing more extensive multi-omic or network-based paradigms to capture the complexity of lncRNA-mediated regulation in pregnancy complications
We are grateful to your thorough and appropriate suggestions bringing to a more tempered subtle and precise assessment of the range and limitation of our study
- Improvement in Analysis Methodology for the Future: Future studies could greatly benefit from multivariate analyses. Methods such as logistic regression would allow evaluating the independent effect of lncRNAs on pregnancy outcomes, adjusting for relevant clinical variables (maternal age, parity, body mass index, other comorbidities). This would provide a more robust understanding of the association. Predictive models using methodologies based on Random Forest could even be addressed. Some comment should be made in the conclusions.
AUTHORS RESPONSE point2:
Dear reviewer, thanks for the helpful and positive input.
We followed your opinion by inserting a separate paragraph in Future Research Directions in the Discussion section: More specifically we now strongly support the need for multivariable analysis methods, e.g., logistic regression to evaluate the independent association of certain lncRNAs with pregnancy outcomes after adjusting for potentially relevant clinical variables such as maternal age, parity, BMI and comorbidities in future studies
We concur, that such approaches could yield more robust insights into lncRNAs and pregnancy disorders, as well as improve the interpretability and clinical relevance of the findings.
Once again thank you for doing so contributing to improved and sharpened "forward-looking" part of the paper
- Minor Issues: An inconsistency in the use of "IncRNA" and "IcnRNA" is observed throughout the text. Ensure that each citation directly supports the information to which it is attached.
AUTHORS RESPONSE 3:
Thank you very much for your thoughtful and useful comment
We agree that the correct acronym is lncRNA (long non-coding RNA), and indeed took great care to conduct a full reading of the manuscript for latent typographical mistakes regarding this abbreviation (such as "IcnRNA") Regardless, we were unable to detect any further appearances of this mistake in the published manuscript as it appears today
We appreciate your attention to detail however and have used this as an opportunity to reconfirm consistency of terminology and citation placement throughout the manuscript to ensure that each reference indeed applies to the text to which it is related.
Reviewer 5 Report
Comments and Suggestions for Authors
In this review the Authors recapitulated the findings of three previously published papers to compare the relationships of the expressions of long non-coding RNA (lncRNAs) with adverse pregnancy outcomes (APOs) in humans. The Reviewer suggests that the three case studies (3 published papers) used in the review do not provide adequate comparative analysis of the subject topic of the systemic review paper, which is marred with several limitations that could affect its scientific value (L275-277). The functional roles of lncRNA is APOs have been extensively explored in the case study papers (Ref. #26-28). For examples, Dai et al. (Ref. # 26) reported that seven lncRNAs were correlated with severity pregnancy-induced hypertension (PIH), whereas Zou et al. 2021 (Ref. #27) reported that the expressions of three lncRNAs were downregulated in patients with intrahepatic cholestasis of pregnancy (ICP). There is nothing new in this review paper, which is mainly based on the summary of the three case study papers.
Author Response
We kindly want to indicate that our work is not an original article but a review. Accordingly, it is not its ambition to introduce new empirical findings but to contribute an intense, systematical and methodologically adequate compilation of previous work according to international standards.
Like you also mentioned, there were only 3 studies that satisfied the inclusion criteria. This low number is not a weakness of our method but indicates the early stage of development in this field. The limited studies themselves are a main finding of the present review, highlighting the need for more research in the role of lncRNAs in APO.
Finally, we would like to stress that this is the first systematic review ever carried out on the subject. Because of the relative newness of the field and the rising popularity of lncRNAs as possible diagnostics and therapeutics, we hope our review will be of help for those entering the area.
Our work is by no means a mere description of the three studies, but instead it takes a critical approach to the methodological way in which these studies were designed and conducted and to the results and limitations we find in them. We also identify crucial challenges, gaps in the literature, and concrete recommendations for future research. This encompasses the requirement for multivariate models, external validation, as well as more comprehensive methods to address the pleiotropic roles played by lncRNAs.
We think this contribution is timely and important. It simplifies the existing understanding and generates new research. In fact, at the time of completing this review article, we had just submitted a large grant proposal (1 milion of euros) to undertake novel multicentre studies of lncRNA and pregnancy outcomes, providing evidence of our personal enthusiasm for furthering this area.
In summary, even though we have presented limited information in this review the scientific interest of the work presented is on the novelty, methodological aspect and the possibility to understand and open the way to future research in this rapidly expanding area of reproductive medicine.
We highly appreciate your comments, for this gave us the possibility to emphasize more the significance and the scientific background of our study.
Round 2
Reviewer 2 Report
Comments and Suggestions for Authors
The authors have provide a rapid response to the reviewers and turnaround of their paper. They have made some changes to the discussion as I requested however I still feel that between lines 246 to 285 the discussion is limited in its scope ie does not consider other studies such as
Zhou S, Li J, Yang W, Xue P, Yin Y, Wang Y, Tian P, Peng H, Jiang H, Xu W, Huang S, Zhang R, Wei F, Sun HX, Zhang J, Zhao L. Noninvasive preeclampsia prediction using plasma cell-free RNA signatures. Am J Obstet Gynecol. 2023 Nov;229(5):553.e1-553.e16. doi: 10.1016/j.ajog.2023.05.015. Epub 2023 May 19. PMID: 37211139.
Ogoyama M, Takahashi H, Suzuki H, Ohkuchi A, Fujiwara H, Takizawa T. Non-Coding RNAs and Prediction of Preeclampsia in the First Trimester of Pregnancy. Cells. 2022 Aug 5;11(15):2428. doi: 10.3390/cells11152428. PMID: 35954272; PMCID: PMC9368389.
These studies also find MALAT1 and H19 and other lncs are reported
I think that providing the alternative names for the lnc in addition to the gene notation ENST for the Dai study would help the reader.
Comments on the Quality of English Language
Lines 246-285 This section of the discussion is confused and contains duplicating non specific statements. I strongly recommend a rewrite or an editors help. The authors also added Table 5 which is also a little rushed and adds little to the paper please reconsider the title of table 5 and I think the authors mean lncRNAs are implicated not interested :)
Author Response
Dear Review,
Thank you for your recommendation. We have added the following papers to our discussion. The first one is particularly interesting, while the second serves as a review. We have incorporated their findings to support our own results.
Zhou S, Li J, Yang W, Xue P, Yin Y, Wang Y, Tian P, Peng H, Jiang H, Xu W, Huang S, Zhang R, Wei F, Sun HX, Zhang J, Zhao L. Noninvasive preeclampsia prediction using plasma cell-free RNA signatures. Am J Obstet Gynecol. 2023 Nov;229(5):553.e1-553.e16. doi: 10.1016/j.ajog.2023.05.015. Epub 2023 May 19. PMID: 37211139.
Ogoyama M, Takahashi H, Suzuki H, Ohkuchi A, Fujiwara H, Takizawa T. Non-Coding RNAs and Prediction of Preeclampsia in the First Trimester of Pregnancy. Cells. 2022 Aug 5;11(15):2428. doi: 10.3390/cells11152428. PMID: 35954272; PMCID: PMC9368389.
We have included alternative names for the lncRNAs where possible.
Thank you for your constructive recommendation and for your excellent work.
Reviewer 5 Report
Comments and Suggestions for Authors
The Authors have addressed my concerns, however, I still reiterate that the number of case studies, based on three published papers, is limited and do not sufficiently provide adequate comparative investigations of the subject topic. I would encourage the Authors that their future case studies (review) should be based on more published papers, and that their contribution to the review should be at least 10-15% of the total cited references, which has been neglected in this study.
Suggestion:
Should consider to provide a table showing the pros and cons of this type of study. Some of the information is given in the text (L268-285), but it will be more explicit when given in the table.
Author Response
Reviewer:
Suggestion:
Should consider to provide a table showing the pros and cons of this type of study. Some of the information is given in the text (L268-285), but it will be more explicit when given in the table.
Author response:
Dear Reviewer, We have included a table expressing this in detail (L268-285).
It tells us that we are writing a more robust scientific paper, and we are all grateful for your input.
We are aware that papers of such nature are few, but this is a very new dimension of studies.
If we inspire the scientific community with our results there may be more to come.
We aim at providing some insights into this topic in order to prompt the generation of new markers.
We have to give (this is the first systematic review) more to provoke, but to compete on the ‘field of medicine’, it is necessary a new content.
Thak you for your hard work.
Round 3
Reviewer 2 Report
Comments and Suggestions for Authors
Have the authors uploaded the correct version of the manuscript V4? The only difference I can see in the manuscript is that table 5 disappeared. This does not correspond to the authors descriptions in their response to reviewers which describe a revised and extended discussion as requested by the reviewer. At present the V4 manuscript has not been substantially revised and a complete rewrite of the discussion is needed. All revisions should be highlighted and there are none in the text of V4.
Comments on the Quality of English LanguagePlease see my comment above